# Gender-Neutral Toilets: A Qualitative Exploration of Inclusive School Environments for Sexuality and Gender Diverse Youth in Western Australia

**DOI:** 10.3390/ijerph191610089

**Published:** 2022-08-15

**Authors:** Jacinta Francis, Pratishtha Sachan, Zoe Waters, Gina Trapp, Natasha Pearce, Sharyn Burns, Ashleigh Lin, Donna Cross

**Affiliations:** 1Telethon Kids Institute, 15 Hospital Avenue, Nedlands, WA 6009, Australia; 2Centre for Child Health Research, The University of Western Australia, 35 Stirling Hwy, Nedlands, WA 6009, Australia; 3School of Psychological Science, The University of Western Australia, 35 Stirling Hwy, Nedlands, WA 6009, Australia; 4School of Population and Global Health, The University of Western Australia, 35 Stirling Hwy, Nedlands, WA 6009, Australia; 5School of Population Health, Curtin University, Kent St., Bentley, WA 6102, Australia

**Keywords:** bathrooms, toilets, transgender, LGBTQ+, schools, bullying, qualitative

## Abstract

School toilets have been identified by sexuality and gender diverse (SGD) students as the least safe spaces in educational institutions. They are sites of verbal, physical and sexual victimisation. Providing gender-neutral toilets in primary and secondary schools may reduce the bullying and victimisation of SGD students, particularly those who are transgender or gender-diverse. This study explored factors influencing the inclusion of gender-neutral toilets in primary and secondary schools in Western Australia. Thirty-four interviews were conducted from May to December 2020 with policy makers or practitioners (*n* = 22) and school staff (*n* = 12) in Perth, Western Australia. Interviews were conducted online and face-to-face using semi-structured interview guides. A thematic analysis of the cross-sectional qualitative data was undertaken. School staff, policy makers, and practitioners identified school toilets as sites of bullying and victimisation of SGD youth and expressed support for gender-neutral toilets as an anti-bullying strategy. Perceived barriers to introducing gender-neutral toilets in schools included financial and spatial costs, building code compliance constraints, resistance from parents and students, privacy and confidentiality concerns, and cultural appropriateness. Including gender-neutral toilets in schools may reduce school-based bullying and victimisation, and improve the mental and physical health of SGD youth.

## 1. Introduction

Sexuality and gender diverse (lesbian, gay, bisexual, transgender, queer, and asexual) students experience more bullying and harassment at school than heterosexual or cisgender (i.e., those whose gender identity matches their sex assigned at birth) youth [1,2]. Sexuality and gender diverse (SGD) youths’ experiences of bullying have included verbal and physical harassment, exclusion, gender policing, demands for sexual favours, and treatment as sexual perpetrators or deviants [3]. Bullying can negatively impact a range of health and academic outcomes in SGD youth, including school attendance, academic performance, psychological well-being, and substance abuse [2,4,5].

School toilets have been identified by SGD students as the least safe spaces within schools [2]. A 2017 survey about the school experiences of 23,001 SGD students in the United States found that almost 43% of students avoided toilets at school due to safety concerns or feelings of discomfort [2]. The term ‘toilet’ is commonly used in some countries (e.g., Australia, United Kingdom) to depict rooms containing hand basins and sanitary hardware designed to collect human waste. While these rooms are frequently referred to as bathrooms in the United States, bathrooms in Australia are distinct from toilets as they contain additional features such as showering facilities and change rooms. Other toilet synonyms used internationally include washrooms, restrooms, and lavatories.

Although SGD students have reported feeling unsafe in school toilets, toilets are particularly hostile environments for trans and gender diverse (henceforth trans) students, and have been linked to reports of verbal, physical, and sexual assault [6,7,8]. While accurate estimates of school-aged trans young people in Australia can be difficult to obtain [9], a national survey of 6327 students enrolled in grades 10, 11, and 12 found that 2.3% of students identified as trans [10]. Another Australian survey of 1407 trans youth aged 14–21 found that over 70% of participants experienced issues regarding toilet access, 60% felt uncomfortable or unsafe accessing toilets, and 39% limited how much they ate or drank in order to avoid public toilets [11]. Further analyses of 922 trans students from the same dataset found that only 29% of students attending secondary schools felt they could safely use the school toilets that matched their gender identity [11]. Similar studies in the United States found 59% of trans adults had avoided toilets at school, work or in public places, with 12% experiencing harassment or assault in toilets, and 31% limiting drinking or eating in order to avoid toilets [6]. The subsequent avoidance of toilets by trans populations may result in dehydration, urinary tract or bladder infection, urinary leakage, and poor concentration [8,12,13]. In addition, many trans students are prevented from using school toilets that reflect their gender identity due to restrictive school policies [2,7,14]. Bathroom discrimination experienced by trans youth aged 13–24 has been associated with depression and suicide attempts [15].

Including gender-neutral toilets in schools may be one strategy to prevent the bullying and assault of SGD students, and related physical and mental health problems. Indeed, many advocates for sexuality and gender diversity have recognised the importance of including gender-neutral toilets in schools and organisations serving SGD youth [8,15,16,17,18,19]. Gender-neutral toilets—also known as all-gender, gender-inclusive, and unisex toilets—are accessible to all students, regardless of their gender identity. While gender-neutral toilets are often discussed in relation to trans youth [18,20], youth of diverse sexuality have also expressed a desire for gender-neutral toilets in schools, identifying them as safe spaces that reflect an inclusive school culture [16]. For example, gender-neutral toilets and signage have been linked to positive sentiments about organisations and feelings of comfort in SGD youth [16]. Sexual minority adults have also supported the inclusion of gender-neutral toilets in public spaces, noting that gender-segregated toilets are often sites where SGD people experience the public gaze, and can be hostile and uncomfortable environments for individuals with diverse sexual orientations [21].

Despite growing support for gender-neutral toilets in the published literature, there can be resistance to the introduction of gender-neutral toilets in schools and public spaces [22,23]. A commentary by Barnett and colleagues (2018) noted that resistance to gender-neutral toilets often reflects a belief that any deviation from the gender assigned at birth is abnormal, and that trans people pose a risk to others and are more likely to commit sexual assaults. Opponents of gender-neutral toilets have also argued that individuals with sexual disorders will disguise themselves as the opposite sex to enter gender-neutral toilets and commit sexual offences [22]. Safety, privacy, and hygiene concerns were also reported by students at a university in Taiwan, although most students did not report any concerns and endorsed gender neutral toilets [24]. Other barriers to introducing gender-neutral toilets include the cost of installing new toilets or renovating existing facilities. While converting disabled or female toilets into gender-neutral facilities is often cheaper than removing urinals in male toilets, this can exacerbate the inadequate provision of existing toilets for disabled and female populations [21,25].

The authors are currently unaware of studies in the Australian or international literature investigating barriers to providing gender-neutral toilets in primary or secondary schools. This paper reports on a key theme from a larger qualitative study exploring school environmental factors influencing bullying behaviour and mental health in primary and secondary school students. Guided by a social-ecological model of human behaviour that recognises the interrelationship between individuals and their built, social, and socio-cultural environments [26], the larger study aimed to identify features of the school built, social, and policy environments that influence bullying and mental health in school students. Barriers to, and facilitators of, designing or modifying the school environment to prevent bullying and improve mental health were also explored. Building on the initial findings of the larger study, the current paper aims to identify factors influencing the inclusion of gender-neutral toilets in primary and secondary school students in Western Australia (WA).

## 2. Materials and Methods

Ethics approval to conduct this qualitative study was granted by the Human Research Ethics Office at The University of Western Australia (RA/4/20/4995), with governance approval granted by Catholic Education WA and the Department of Education WA. A qualitative study design was chosen for this study because the impact of gender-neutral toilets on bullying behaviours involving SGD youth is a relatively new area of research. As such, it is well suited to qualitative research methods, such as interviews, that allow researchers to investigate complex phenomena and explore participant responses [27]. Given that discussions about bullying, mental health, sexuality, and gender diversity are potentially sensitive topics, semi-structured one-on-one interviews were chosen for this study to protect the privacy and confidentiality of study participants and to ensure rich, inductive data.

### 2.1. Sampling and Recruitment

Policy makers and practitioners were invited to participate in individual interviews, between May and December 2020. Practitioners included school architects, private consultants (for example, from disability services or landscape and playground design), and academics specialising in bullying prevention, education, mental health, and the health of Aboriginal and Torres Strait Islander people. Policy makers and practitioners (hereafter referred to as policy makers) were recruited using a snowball sampling technique, after consultation with the research team’s Stakeholder Advisory Committee. The Committee comprised representatives from WA school sectors, government departments, school associations, private architectural firms, tertiary education institutions, and child health research institutes.

Staff from Perth metropolitan schools were recruited and interviewed between August and December 2020. Although the commencement of the COVID-19 pandemic in March 2020 delayed the recruitment of school staff and students, it is unlikely to have impacted participants’ responses regarding gender-neutral toilets and SGD youth. Students in WA were only required to move to online learning platforms for approximately two weeks in April 2020 before returning to school. Primary schools in WA typically include grades kindergarten to 6 (students aged 4 to 12), while secondary schools include grades 7 to 12 (students aged 12 to 18). Maximum variation sampling—a purposive sampling technique that captures a wide range of perspectives—was used to recruit schools from different school sectors, socio-economic areas, years since establishment, and school levels (i.e., primary or secondary). Six schools were recruited, with schools evenly divided between (i) education sectors: Independent, Catholic, and Government; (ii) primary and secondary; (iii) low and high socio-economic areas; and (iv) old and new school builds, with “old” schools built prior to 1979. Socio-economic status was determined using the Index of Community Socio-Educational Advantage (ICSEA) [28]. ICSEA scores < 1000 represented low socio-economic schools and scores ≥ 1000 represented high socio-economic schools.

School principals were contacted via email by the research team to ascertain their interest in the study. Consenting principals nominated a study coordinator to assist the research team in recruiting school staff working with students in grades 4 to 10. Staff included, but were not limited to, principals and deputy principals, classroom teachers, specialist subject teachers, chaplains, and school psychologists. Policy makers, school principals, study coordinators, and study participants received a participant information and consent form explaining the purpose of the study and requirements of participation. All participants provided written consent to participate in the study prior to the interviews.

### 2.2. Data Collection

A semi-structured interview discussion guide was developed by a research team with expertise in school design, built environment, bullying prevention, mental health, and school-based interventions. The interview discussion guide was piloted with a convenience sample of three adults for comprehension, resulting in minor changes to the guide. Participants were asked to identify (i) individual, built, social, and policy factors influencing bullying behaviour in school students; (ii) changes to the school environment to prevent bullying; and (iii) barriers to and enablers of suggested changes. The impact of gender-neutral toilets on SGD students was an unanticipated theme that emerged during analyses. Bullying was defined at the commencement of the interview as the repeated, intentional harm of a person who has less power than the aggressor [29].

All interviews were conducted by the first author, an experienced qualitative researcher, skilled in interview and focus group techniques. Study participants were given the option to be interviewed via videoconference, telephone, or face-to-face. Face-to-face interviews were conducted at the participant’s workplace or school.

Interviews with policy makers ranged between 40 and 70 min, and with school staff between 20 and 45 min. Interviews were audio-recorded and transcribed verbatim. Data collection continued until no new codes or themes emerged from the data [30,31]. A survey designed to capture demographic data was self-administered at face-to-face interviews, and administered by the interviewer during videoconference interviews.

### 2.3. Data Analysis

To maintain dependability and determine credibility, interview transcripts were checked by two researchers for accuracy, and analysed with the assistance of qualitative research software package, QSR NVivo version 12. A theoretical thematic analysis was conducted, with a coding framework based on individual, social environment, built environment, and policy factors underpinning the project’s social-ecological framework, as well as the study aims and interview discussion guide.

Analysis was guided by Braun and Clarke’s (2006) framework for thematic analysis [32]. Two research team members familiarised themselves with the data by re-reading transcripts and developing initial ideas and codes. Open coding was used to assign the data to codes, which were then grouped into potential categories and themes aligning with the project’s social-ecological framework. Categories were clustered further into overarching themes associated with the school’s built environment and bullying. Themes, categories, and codes were confirmed by a second researcher to maintain confirmability and enhance dependability [33]. Built environmental factors influencing bullying behaviour were a theme in the larger study, with toilets identified as a sub-theme and gender-neutral toilets identified as a category. In the current study, gender-neutral toilets comprised the main theme, after categories and codes specific to gender-neutral toilets were identified. To reduce bias and enhance confirmability, the coding and themes were discussed with the research team, with minor inconsistencies between researchers discussed until a consensus was reached. The Consolidated Criteria for Reporting Qualitative Research (COREQ) checklist guided the reporting of this study [34].

## 3. Results

Table 1 presents the characteristics of study participants. All policy makers and school staff identified as cisgender. Of the 26 policy makers invited to participate in the study, 22 consented. Of the 12 schools invited to participate in the study, six schools consented.

Codes falling under the theme of gender-neutral toilets included the bullying experiences of SGD students in toilets, support for gender-neutral toilets, and specific barriers to including gender-neutral toilets in schools, such as financial and spatial costs, building code compliance constraints, resistance from parents and students, privacy and confidentiality concerns, and cultural appropriateness (Table 2).

### 3.1. Bullying Experiences of SGD Students in Toilets

School toilets were identified by study participants as potential bullying hotspots for trans students:


*We’ve got one young man who’s gone from man to woman…when he was going into the boys’ toilets, he was [verbally assaulted]. And then when he first started going into the girls’ toilets as his new identity, he was getting some of them being a bit nasty to him.*
(School staff #09)

Study participants identified perceived increases in the number of trans students experiencing problems in school toilets. When asked if gender identity impacted students’ use of school toilets, one staff member noted that a growing number of students were identifying as trans:


*[Toilet provision for trans students is] a big issue…I’m only learning about that now because we’ve only come against it in the last two years.*
(School staff #09)

Similarly, policy makers reported an increase in requests by school principals and staff for gender-neutral toilets:


*There is an increasing number of requests: “we need a toilet, we’ve got a trans child, we need a separate toilet”.*
(Policy maker #19)

Participants also recognised, however, that the prevalence of bullying victimisation in trans students can be difficult to determine. One policy maker referred to the findings of a recent report on student safety, noting that students were reluctant to report incidents to staff for fear the disclosure would lead to further harassment:


*[Trans students] were often bullied at school and they didn’t disclose the bullying because it might lead to a disclosure about their gender identity, which would stigmatise them or make them more of a subject of bullying. So they felt like they had to keep a double secret.*
(Policy maker #09)

### 3.2. Support for Gender-Neutral Toilets

Policy makers and school staff expressed support for gender-neutral toilets, particularly as a strategy to reduce bullying of trans students:


*I’ve seen that there’s been this development of male toilets, female toilets, and then a unisex toilet. So there’s a choice. If there’s no choice, then you’re stuck with a binary issue which, of course, some kids feel challenged by. And for some transgender [females]… they may well feel quite victimised going into a male toilet because that’s probably a site of bullying.*
(Policy maker #09)

Participants also noted other benefits of gender-neutral toilets, including their potential to eliminate self-consciousness experienced by trans students. For example, one policy maker identified the internal dialogue that may occur within a trans student when visiting a disabled toilet instead of a male or female toilet:


*“Look at me, I have to go to that disabled toilet to go to the toilet, and I’m a boy, but I’m identifying as a girl, and everyone will be going errh?” The way of having the gender-free toilets is that it doesn’t matter which one you go to.*
(Policy maker #19)

Cultural changes within schools were also reported by study participants, with staff expressing a growing acceptance of gender-neutral toilets:


*I used to think it was okay to have the disabled toilet, but my principal corrected me the other week and said, “Absolutely not, they should be able to go into the toilet of the gender of which they identify”.*
(School staff #09)

While some gender-neutral toilets allow students to occupy a shared space with multiple cubicles, other gender-neutral toilets are single, self-contained spaces with a toilet and hand-wash basin. The potential benefits of self-contained toilets were expressed by the following policy maker:


*[Students] can just use it so you don’t have to actually deal with the gender issue… if you can actually have a controlled space and this is your space and you are doing that personal thing, then you feel more comfortable anyway, which takes away that fear. There’s also no opportunity for bullying because it’s a single space for a single person.*
(Policy maker #01)

### 3.3. Barriers to Incorporating Gender-Neutral Toilets into Schools

Although many policy makers and school staff supported the inclusion of gender-neutral toilets in schools, participants acknowledged resistance from some members of the school community, including governing bodies. Reasons for this resistance varied, with the most common barriers including financial and spatial constraints, building code compliance constraints, resistance from parents and students, privacy and confidentiality concerns, and cultural appropriateness.

#### 3.3.1. Financial and Spatial Costs of Gender-Neutral Toilets

Although participants noted the bullying prevention benefits of self-contained toilets, the financial cost of renovating or building such toilets was a common theme identified by study participants:


*The problem with [a gender-neutral self-contained toilet] is it can be [expensive] as well, cost more money and so forth.*
(Policy maker #01)

The spatial costs of incorporating self-contained toilets were also noted:


*Gender-neutral toilets can be done a number of different ways, and the one that is about making a series of completely self-contained cubicles that have hand washing as well, that is the one that immediately takes more space than would currently be briefed, and therefore, would sort of tend not to be the first cab off the rank for an approach.*
(Policy maker #20)

As the above quote illustrates, policies that dictate spatial constraints can also be barriers to incorporating gender-neutral self-contained toilets.

#### 3.3.2. Building Code Compliance Constraints

Adherence to the National Construction Code (NCC) was identified repeatedly as a barrier to the inclusion of gender-neutral toilets in Australian schools. The NCC is a set of technical design and construction provisions for buildings in Australia that sets minimum requirements for the safety, health, amenity, accessibility, and sustainability of buildings [35]. Policy makers noted that the NCC contains stipulations about the number of male and female toilets on school sites:


*There are also some code compliance issues around the gender-neutral toilets because under the National Construction Code there are actually a stipulated number of female, special or male toilets that have to be provided and it doesn’t accommodate things like non-gender.*
(Policy maker #11)

Other comments by policy makers indicated that school designs that did not adhere to the NCC required expensive consultations before approval could be granted:


*Anyone who…builds a school…has to follow the codes for that particular type of building… It requires extra consultants to come and design something specifically and tick it off. And that consultant takes responsibility for their design and so it’s not that easy. It’s not cheap and it’s not ideal. We would love to see the NCC changed to something where gender-neutral is recognised easily.*
(Policy maker #12)

However, one participant offered a potential solution to including gender-neutral toilets in schools while still adhering to the NCC:


*We actually require a lot more toilets than is required under the code, so what we have done in the case of this school I am talking about is the ones which are required to be under the National Construction Code are gendered and all the ones we are not required to provide…we are doing those as gender neutral. And I think that is a really good outcome because we meet code-compliance and we give choice… we will see how it works as a little pilot project and see what reaction we get from students and teachers.*
(Policy maker #11)

#### 3.3.3. Resistance from Parents and Students

Participants noted that conversations about gender-neutral toilets were often met with resistance from some students and parents:


*I can’t see us putting in gender-neutral toilets into a primary school. I don’t know how that would work. It would be great but also with challenges. Parents wouldn’t allow it.*
(Policy maker #12)

Our findings also demonstrated potential resistance from parents of trans students who did not agree with the design of gender-neutral toilets available to students. While some parents of trans youth were content with schools offering their children a gender-neutral self-contained toilet, other parents were concerned these toilets would increase their child’s risk of victimisation:


*At the moment we’re still going with male/female toilets, but then having a number of toilets that are gender-neutral that can be used… obviously, you get two lots of parents’ thinking; [some] want [contained gender-neutral] toilets, and then other parents don’t want their students going to that special toilet… I think it’s just fear of the unknown. …we have parents that don’t even like the students having to use the AATs [Ambulant & Accessible Toilets], which are the student accessible toilets, because it fingers their child as being different… People are worried about their children and what other kids will say about them.*
(Policy maker #21)

#### 3.3.4. Privacy and Confidentiality Concerns

Concerns about privacy and confidentiality were also identified as barriers to including gender-neutral toilets in schools. As noted earlier in the study findings, some trans students were reluctant to report bullying incidents for fear that it would disclose their gender identity. Concerns about privacy were also relevant to cisgender students accessing shared toilet stalls:


*The school was quite concerned—it’s a high school—about the privacy of the student in the cubicle, and particularly worried about mobile phones and someone lifting a phone up over the top and taking a photo of a kid on the toilet, or under the door and just doing those intimidating things…Instead of doing cubicles we did what we call a super loo, which is where it’s like a home. You have a toilet and a hand basin in a room with a door in a frame with a lock, so you go into the room and you are completely private.*
(Policy maker #02)

Participants also noted, however, that self-contained toilets with floor-to-ceiling walls presented other challenges for staff and students. While self-contained toilets offer students more privacy, they also make it more difficult to supervise and identify problematic behaviours, such as smoking or vaping, self-harm, or the victimisation of students who have been corralled into self-contained toilets against their will.

#### 3.3.5. Cultural Appropriateness of Mixing Genders

The cultural appropriateness of gender-neutral toilets was also discussed by study participants:


*We are also aware that there is a cultural issue if you suddenly start putting non-binary toilets everywhere. There’s some cultures where that mixing of male and female is actually seen to be really very culturally inappropriate…Muslim and Islamic faith and some parts of Indigenous cultures…There’s quite strict cultural rules around that so it’s not a matter of saying we are just going to do everything like this.*
(Policy maker #11)

## 4. Discussion

This study appears to be the first exploration of factors influencing the inclusion of gender-neutral toilets in Australian schools. School toilets were identified by policy makers and school staff as common locations of bullying and victimisation of trans youth. While participants supported gender-neutral toilets as an anti-bullying strategy, barriers to including gender-neutral toilets in schools included financial and spatial costs, building code compliance constraints, resistance from parents and students, privacy and confidentiality concerns, and cultural appropriateness.

The findings that trans students are bullied in school toilets in Western Australia reflect reports from the United States showing that school toilets are common sites of harassment and victimisation of trans youth [2,6,7]. Participants in this study spoke almost exclusively about trans students when discussing gender-neutral toilets. This emphasis on gender minorities may suggest that gender-segregated toilets are less problematic for sexuality-diverse youth than trans youth. However, it may also support studies showing trans youth experience more victimisation through bullying than their sexuality-diverse counterparts, or that policy makers and school staff are unaware of the challenges faced by youth of diverse sexuality when accessing school toilets [36]. Indeed, studies have shown that youth of diverse sexuality often feel unsafe in gender-segregated toilets and associate gender-neutral toilets with feelings of comfort and inclusivity [16,21]. Further investigations into the toilet experiences and preferences of sexuality-diverse students are needed.

The current study identified two types of gender-neutral toilets: self-contained toilets accessed by one person at a time, and multi-stall toilets that can accommodate multiple people. Barriers to including gender-neutral toilets in schools may vary according to toilet designs. For example, while self-contained toilets may be restricted by financial and spatial constraints, multi-stall toilets may not accommodate the needs of religious or cultural minority groups. Study participants noted that mixing students of different genders may not be acceptable to people from Australian Indigenous cultures or Muslim and Islamic faiths. Other researchers have also noted that Islamic, Hindu, and Orthodox Jewish religions often do not permit females to share public toilets with male strangers, particularly when menstruating [25]. In Kuwait, gender segregation is not only enforced in public toilets, but also in public schools, government institutions, and public transportation [37]. Future studies might explore different religious and cultural perceptions of gender-neutral toilets, including the perceptions and experiences of SGD students from different cultural backgrounds. The experiences of SGD students attending single-sex schools also warrant further investigation, given that trans students attending single-sex schools have reported challenges around restrictive toilet access, enrolment barriers, gendered language, restrictive gender norms, uniforms, and change-room facilities [8].

By identifying barriers to gender-neutral toilets in schools, this study has identified key areas requiring intervention. For example, study participants pointed to the current National Compliance Code (NCC) that guides the design and construction of buildings in Australia and does not accommodate gender-neutral toilets. Changes to the NCC require the submission of a Proposal for Change to the Australian Building Codes Board [35]. However, despite the exclusion of gender-neutral toilets from the NCC, some states in Australia have introduced policies that support trans students in schools, including a mandated procedure from the Department for Education in South Australia to support trans and intersex children and young people. The procedure states that trans students should be able to use facilities that match their gender identity, and that alternative options, such as using disabled or staff facilities, should be temporary solutions unless requested by the student [38]. Gender-neutral toilets became a debated topic in the United States around 2017 when President Donald Trump revoked instructions by the 2016 President, Barack Obama, that allowed students in the United States to access school toilets matching their gender identity [39]. President Joe Biden’s administration reversed Donald Trump’s decision in 2021, although it is still unclear how this decision applies to school toilet access [40]. Intervention studies exploring the positive effects of gender-neutral toilets on SGD youth will not only strengthen submissions to change existing building codes in Australia, but may also support advocates of SGD youth and gender-neutral toilets internationally.

While participants in the current study supported the introduction of gender-neutral toilets in schools, they also noted that resistance from parents may be stronger in primary than secondary schools. Indeed, parents of children attending a primary school in London petitioned against the inclusion of gender-neutral toilets over concerns of increased sexual assaults [23]. Analyses of criminal incident reports have found no evidence that assaults, sex crimes, or voyeurism in public toilets are increased by gender-neutral toilets or SGD inclusive policies [41]. Furthermore, trans youth are more likely to be victims of sexual assault than perpetrators [7]. Gender-neutral toilets may also be safer for women than gender-segregated toilets, because of higher volumes of users and enhanced natural surveillance [24,42]. Conducting trials of gender-neutral toilets in secondary schools may be more acceptable to parents, allowing researchers to explore the needs of SGD students before commencing trials in primary schools. Using terms such as ‘universal access toilets’ rather than gender-neutral, all-gender, or unisex toilets may also reduce resistance. Eliminating resistance from parents and students may also require changes to school culture, such as encouraging school administrators to foster inclusive school climates, supporting the development of gay–straight alliance groups, introducing and promoting SGD-specific anti-bullying policies, encouraging the use of preferred names and pronouns, removing gendered uniform codes, and including examples in the school curriculum that normalise SGD identity [11,16,43,44].

Choosing the location of gender-neutral toilets in schools also requires careful consideration. Locating gender-neutral toilets in isolated and remote areas of the school has the potential to increase the stigmatisation and insecurity experienced by some SGD youth [16]. Similarly, retrofitting disabled or female toilets as gender-neutral toilets can cause issues if facilities serving disabled and female populations are already lacking [21,45]. Furthermore, many trans youth and adults resist the implication that being part of a gender minority is equivalent to a disability [21,46]. Participants in this study noted that providing students with a range of toilet designs and access options may be the best approach when incorporating gender-neutral toilets into schools. Researchers exploring gender-neutral toilets in universities and public spaces have also suggested that societies need to accommodate the co-existence of both gender-neutral and gender-segregated toilets [24,25].

While a qualitative research design is well-suited to studies exploring relatively new areas of research investigating complex phenomena, a limitation is that its study findings cannot be generalised to broader populations. The lack of SGD student voices is also a limitation of this study. While this study has identified barriers to including gender-neutral toilets from the policy-maker, practitioner, and school staff perspective, the opinions of SGD students and adults are crucial when designing inclusive school environments. For example, some quotes in this manuscript demonstrated a poor awareness of appropriate gender pronouns by school staff and policy makers: “We’ve got one young man who’s gone from man to woman…when he was going into the boys’ toilets, he was [verbally assaulted]” and one cisgender adult participant speaking in the voice of a trans person “I’m a boy, but I’m identifying as a girl”. A poor awareness of gender pronouns can increase the likelihood that non-binary youth will be misgendered and victimised [8]. While participants’ mean age reflected their extensive work experience, this study did not include policy makers or school staff younger than 30 years of age. A broader range of perspectives may strengthen future studies. Similarly, barriers to including gender-neutral toilets in schools represented an unanticipated theme emerging during interviews exploring the impact of schools’ built environment on the bullying behaviour and mental health of school students. As such, the findings of this study may reflect the perspectives of participants who had an existing awareness of issues surrounding gender-neutral toilets and SGD youth. However, this study sought the perspectives of participants representing a range of school sectors, socio-economic areas, and school levels, with policy makers and school staff able to identify barriers to introducing gender-neutral toilets that students may not be aware of, such as financial, spatial, and building code compliance constraints.

## 5. Conclusions

This study has demonstrated a growing awareness of the need for gender-neutral toilets in schools. Potential barriers, and in turn, solutions to support the inclusion of gender-neutral toilets in schools have been identified. While gender-neutral toilets may supplement school-based anti-bullying efforts to improve the mental and physical health of SGD youth, changes to school culture that address broader issues of inequality and inclusivity are also required to support the physical and mental health of SGD staff and students.

## Figures and Tables

**Table 1 ijerph-19-10089-t001:** Characteristics of study participants.

Characteristic	Policy Makers (*n* = 22)		School Staff (*n* = 12)	
	*n*	%	*n*	%
**Age (years)**				
31–40	2	9.1	6	50.0
41–50	8	36.4	9	75.0
51–60	6	27.3	3	25.0
61–70	6	27.3	-	-
Mean (SD)	53.3 (8.7)		47.2 (7.7)	
**Gender**				
Male	10	45.5	5	41.7
Female	12	54.6	7	58.3
**Born in Australia**				
Yes	15	68.2	9	75.0
No	7	31.8	3	25.0
**Years in role**				
1–10	2	9.1	3	25.0
11–20	8	36.4	5	41.7
21–30	9	40.9	3	25.0
31–40	2	9.1	1	8.4
41–50	1	4.5	-	-
Mean (SD)	23.6 (18.1)		9.0 (9.1)	

**Table 2 ijerph-19-10089-t002:** Gender-neutral toilets: Theme, Categories and Codes.

Theme	Category	Code
Gender-neutral toilets	Need for gender-neutral toilets	Bullying experiences of SGD students in toilets
		Support for gender-neutral toilets
	Barriers to including gender-neutral toilets in schools	Financial costs
		Lack of space
		Building code compliance constraints
		Resistance from parents
		Resistance from students
		Privacy and confidentiality concerns
		Cultural appropriateness

## Data Availability

Data are available upon reasonable request from the first author.

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
