# Peer review of "Gender-Neutral Toilets: A Qualitative Exploration of Inclusive School Environments for Sexuality and Gender Diverse Youth in Western Australia"

_ijerph, 2022, doi:10.3390/ijerph191610089_

Round 1
Reviewer 1 Report
This is an excellent paper that addresses a crucial topic regarding gender-minority bullying in schools. The paper is clear, well written and makes an important contribution to the field. It will be of immense interest to scholars.
The paper is suitable for publication once the following minor revisions are addressed.
1. A research question is needed at the end of the background section, and/or a better explanation of the larger project.
2. There needs to be a rationale for why qualitative research methods were employed in the methods section rather than at the end of the paper in the limitations section
3. Some reflection on the research ethics employed given the topic area
4. Sampling and recruitment –
An account of the ethnicity of schools and participants is required especially given the discussion.
5. 'Data collection continued until theoretical saturation was achieved' (Glaser and 153 Strauss, 1967) - 'theoretical saturation' is a now a contested concept and the authors need to show evidence of their awareness of this contemporary debate in their rationale for using saturation.
6. There is a lack of gender theory in the described approach of the study. The project's social-ecological framework, as well as the study aims, need to be elaborated upon. This theory needs to be evident in the description of the interview guide, and data analysis.
7. There is no ethnicity in the sample reporting.
8. In the discussion, the authors make the important point that future studies might explore different religious and cultural perceptions of gender-neutral toilets, It is therefore important to know the ethnic/cultural background of the schools and participants in the study.
Reviewer 2 Report
The manuscript “Gender-neutral toilets: Designing inclusive school environments for sexuality and gender diverse youth” is crucial in the context of WASH and gender. Building on the acknowledgement of sanitation access as a human right, the global Sustainable Development Goals (SDGs) have made it an objective and priority to ensure that everyone has access to adequate and equitable sanitation and hygiene by 2030. The manuscript is well written. However, I have the following minor comments.
Title: Please add study settings (Western Australia) and study design (qualitative study/ policymakers’ perspective); otherwise, the title seems like a review study.
Abstract: No comments
Introduction:
Provide the information on an approximate number of transgender populations in a school (if data is available).
The paper forms part of a larger qualitative study exploring school environmental factors influencing bullying behaviour and mental health in primary and secondary school students. Mention what is the objective of this particular study.
Methods:
A thick description of study settings is missing.
How many potential participants have you contacted, and how many have participated?
Please mention the reporting guideline you follow for your study (SRQR/COREQ). \
Results:
Add one Table on the coding tree (theme, category and codes).
The overall results have less description with more quotes.
Discussion
Add one paragraph on ‘Implication for policy and practice’.
Methodological consideration is missing (strength and limitation of the study), including investigator and source triangulation, members checking, peer debriefing etc.
Reviewer 3 Report
The current paper utilized qualitative interviews with policymakers or practitioners and school staff to understand barriers to providing gender-neutral bathrooms in primary and secondary schools. The findings align with cultural narratives of barriers (e.g., parents’ fears for children) and identify practical issues (e.g., regulations about the number of male/female bathrooms). The authors did an excellent job of discussing these barriers and providing possible solutions to the barriers. Overall, I enjoyed reading the paper and believe it makes important contributions to research on gender-neutral bathrooms in primary and secondary schools. I have a few minor comments that the authors need to address.
· The sentence, “Experiences of bullying by sexuality and gender diverse (SGD) youth have included verbal and physical harassment, exclusion, gender policing, demands for sexual favours, and treatment as sexual perpetrators or deviants [3].” Should be re-written to say “Sexuality and gender diverse (SGD) youths’ experiences of bullying have included...” As it is currently written, it sounds as if the SGD youth are doing the bullying.
· The authors should be a bit more reserved regarding their claim towards the end of the introduction that no international studies have examined barriers to providing gender-neutral toilets in primary/secondary schools.
· The authors state, “The impact of gender-neutral toilets on SGD students was an unanticipated theme that emerged during analyses.” While it is great that such an important theme emerged from the data, the authors need to mention in the limitations that this wasn’t necessarily something meant to be captured by the study. The policymakers/staff that mention it were probably more aware of the issues than those who did not mention it. If all the participants mention the issue, this could be stated too.
· The data for the study were collected between May and December 2020. So, the authors need to say something about the COVID-19 context in Australia and how this is (or is not) important for the context of the study. For example, I imagine there was some movement to distance learning, so I wonder how recently these issues came up for the participants.
